# BROADWAY: BOOST YOUR TEXT-TO-VIDEO GENERATION MODEL IN A TRAINING-FREE WAY

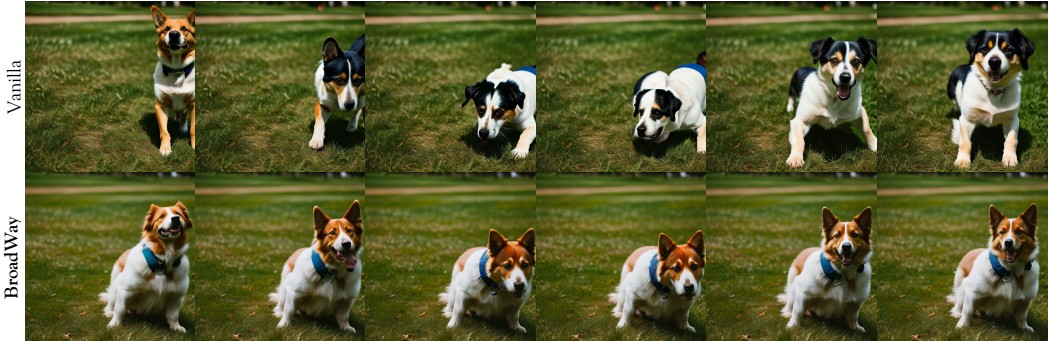

*Prompt: A dog, playing on the grass, soft lightening, high quality, ...*  Consistency ↑

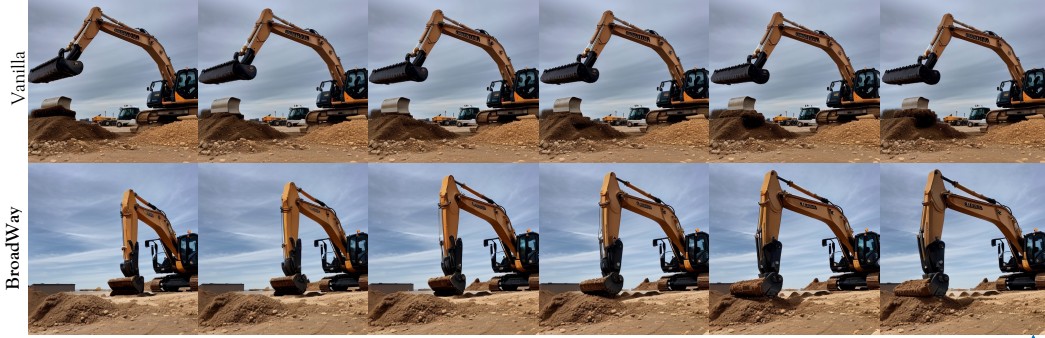

*Prompt: An excavator is digging at a construction site, film grain, ...*  Motion ↑

Figure 1: Given a diffusion based text-to-video (T2V) backbone, **BroadWay** can improve its synthesis quality in a training-free and plug-and-play manner, enhancing both the temporal consistency and the motion magnitude in the sampled results.

## ABSTRACT

The text-to-video (T2V) generation models, offering convenient visual creation, have recently garnered increasing attention. Despite their substantial potential, the generated videos may present artifacts, including structural implausibility, temporal inconsistency, and a lack of motion, often resulting in near-static video. In this work, we have identified a correlation between the disparity of temporal attention maps across different blocks and the occurrence of temporal inconsistencies. Additionally, we have observed that the energy contained within the temporal attention maps is directly related to the magnitude of motion amplitude in the generated videos. Based on these observations, we present **BroadWay**, a training-free method to improve the quality of text-to-video generation without introducing additional parameters, augmenting memory or sampling time. Specifically, BroadWay is composed of two principal components: 1) Temporal Self-Guidance improves the structural plausibility and temporal consistency of generated videos by reducing the disparity between the temporal attention maps across various decoder blocks. 2) Fourier-based Motion Enhancement enhances the magnitude and richness of motion by amplifying the energy of the map. Extensive experiments demonstrate that BroadWay significantly improves the quality of text-to-video generation with negligible additional cost.

# 1 INTRODUCTION

In recent years, the field has observed substantial progress in the evolution of diffusion-based models specifically dedicated to video generation tasks, notably in text-to-video synthesis (Khachatryan et al., 2023; Blattmann et al., 2023b; Guo et al., 2023b; Chen et al., 2024). Despite these advancements, the practical applicability of generated videos remains limited due to inadequate quality. This suboptimal performance is characterized by two predominant issues: firstly, a portion of the generated videos exhibit structurally implausible and temporally inconsistent artifacts, and secondly, another subset of the generated videos demonstrates markedly restricted motion, bordering on the static nature of a still image. Prior methodologies have primarily concentrated on enhancing video generation quality through advances in training mechanisms, such as improving the quality of training data (Blattmann et al., 2023a), scaling training data (Wang et al., 2024b), refining model architecture (Hong et al., 2022) and training strategies (Chen et al., 2024). However, these approaches often entail substantial costs. This work endeavors to improve video generation quality in the inference phase, specifically in the realm of text-to-video generation, without necessitating training, introducing additional parameters, augmenting memory or sampling time.

In current video generation models, an encoder-decoder architecture (Ronneberger et al., 2015) is typically utilized, wherein the decoder is comprised of multiple blocks. Each block integrates several temporal attention modules (Guo et al., 2023b), facilitating the modeling of motion within the generated videos. We have two observations about the temporal attention module. The first is a correlation between artifact presence and the inter-block divergence of temporal attention maps. Specifically, video generation processes exhibiting structurally implausible and temporally inconsistent

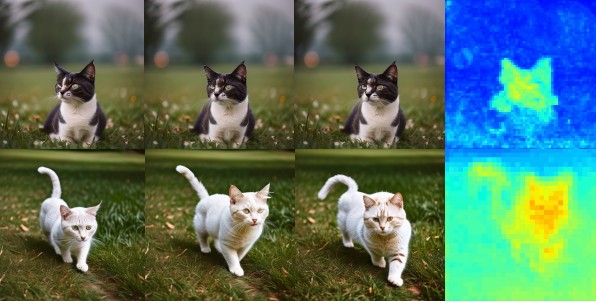

Figure 2: Generated videos with richer motion typically exhibit a higher energy.

artifacts demonstrate greater disparity between the temporal attention maps of different decoder blocks. Conversely, processes devoid of such evident artifacts exhibit reduced disparity among these maps, as illustrated in Fig. 3(a). The second is a correlation between the amplitude of motion in generated videos and the energy of the corresponding temporal attention maps, defined in the method section. Specifically, videos that exhibit a higher degree of motion amplitude and a richer variety of motion patterns are observed to possess greater energy within their temporal attention maps, as illustrated in Fig. 2.

Based on the observations, we present BroadWay, a training-free approach with negligible additional cost to improve the generation quality of T2V diffusion models. BroadWay is composed of two principal components: Temporal Self-Guidance and Fourier-based Motion Enhancement, both meticulously engineered to refine the temporal attention module within T2V models. Temporal Self-Guidance leverages the temporal attention map from the preceding block to inform and regulate that of the current block. This approach effectively mitigates the disparity between the temporal attention maps across various decoder blocks, thereby normalizing their disparity. As a result, videos that initially exhibit structural implausibility and temporal inconsistency, significantly reduce such artifacts through the application of Temporal Self-Guidance, as shown in the first and second rows in Fig. 1. Furthermore, Fourier-based Motion Enhancement modulates the high-frequency components of the temporal attention map, thereby amplifying the energy of the map, as detailed in the methodology section. This enhancement circumvents the generation of videos that closely resemble static image. With the Fourier-based Motion Enhancement, videos that were previously characterized by minimal motion exhibit an increased amplitude and a more diverse range of motion patterns, as illustrated in the third and last rows in Fig. 1.

We evaluate the performance of BroadWay on various popular T2V backbones, including those with additional motion modules trained from frozen T2I models and those trained end-to-end directly for T2V tasks. Our experiments show promising results, demonstrating the effectiveness and strong adaptability of BroadWay. Furthermore, additional experiments reveal that BroadWay also

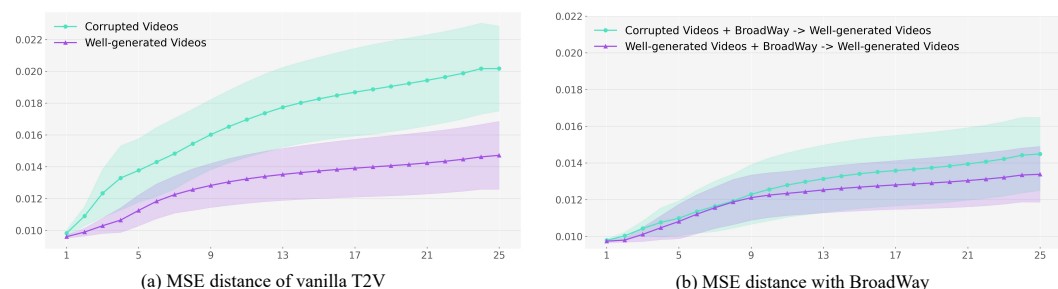

(a) MSE distance of vanilla T2V          (b) MSE distance with BroadWay

Figure 3: MSE distance between temporal attention maps of different levels in UNet.

exhibits potential in the image-to-video (I2V) domain, further expanding the applicability of Broad-Way across various video generation tasks.

Our contributions are summarized: (1) We conduct a deeper analysis of the temporal attention module widely adopted in current T2V backbones, and observe two correlations between the generated videos and corresponding temporal attention maps. (2) We propose BroadWay, which significantly improves the quality of text-to-video generation without necessitating training, introducing additional parameters, augmenting memory or sampling time. (3) BroadWay can be seamlessly integrated with various mainstream open-source T2V backbones like AnimateDiff and VideoCrafter2, demonstrating strong applicability and extensibility.

## 2 RELATED WORK

### 2.1 TEXT-TO-VIDEO DIFFUSION MODELS

Given a textual prompt, text-to-video (T2V) diffusion models (Singer et al., 2022; Hong et al., 2022; Wang et al., 2023b; Chen et al., 2023; Wang et al., 2023a; 2024a; Khachatryan et al., 2023) aim to synthesize image sequences that maintain both temporal consistency and textual alignment. Unlike text-to-image (Ding et al., 2021; Zeqiang et al., 2023; Saharia et al., 2022; Podell et al., 2023) that emphasizes perfecting individual images, T2V poses a heightened challenge of maintaining both visual aesthetics for each frame and the realistic motion between frames. To this end, most approaches incorporate extra motion modeling modules into existing image diffusion architecture, leveraging the underlying image priors. For instance, AnimateDiff (Guo et al., 2023b) introduced trainable temporal attention layers to frozen text-to-image models to effectively capture the frame-to-frame correlations Some works (Blattmann et al., 2023b; Chen et al., 2024) combined temporal convolution modules and temporal attention layers for modeling short/long range dependencies. To alleviate motion synthesis difficulty, Ge et al. (Ge et al., 2023) suggested employing temporally related noise to enhance temporally consistent. Nevertheless, due to the scarcity of high-quality video data and the intricacies of motion synthesis, the current available T2V models still struggle to harmonize motion strength with motion consistency. This work identifies that the consistency across temporal attention blocks indicates the continuity of synthesized video sequences while the energy within the temporal attention maps dominates the magnitude of motion, and thus proposes a training-free strategy to unlock the potential of exiting T2V models by encouraging uniform motion modeling and enhanced frequency energy.

### 2.2 DIFFUSION FEATURE CONTROL

Controlling targeted diffusion features to manipulate specific attributes has been demonstrated to be an effective strategy in the realm of image and video synthesis (Chefer et al., 2023; Kim et al., 2023; Xiao et al., 2023; Liu et al., 2023; Qi et al., 2023),. Prompt2Prompt (Hertz et al., 2022) revealed that the cross attention maps domain the image layout. DSG (Yang et al., 2024) proposed that spatial means of diffusion features represent the appearance, which offers simple approach for image property manipulation, such as size, shape, and location. FreeControl (Mo et al., 2023) suggested to perform image structure guidance by aligning the PCA features with given reference image in spatial self-attention block, providing a versatile counterpart of ControlNet (Zhang et al., 2023). DIFT (Tang et al., 2023) observed that the semantic corresponding can be directly extracted by spatially

measuring the difference between diffusion feature. MotionClone (Ling et al., 2024) demonstrates the sparse control of temporal attention maps facilitates a training-free motion transfer, enabling reference-based video generation. FreeU (Si et al., 2024) suggested re-weighting the contribution of skip features and backbone features by using spectral modulation and structure-related scaling, promoting the emphasis on backbone semantics. In this work, we propose Temporal Self-Guidance to facilitates uniform motion modeling across blocks by narrowing the disparities between temporal attention maps. This is work together with Fourier-based Motion Enhancement, which boosts motion magnitude by amplifying frequency energy, thus elevating the quality of the generated videos

## 3 PRELIMINARY

### 3.1 LATENT DIFFUSION MODEL

In the context of T2V generation, latent diffusion model (Rombach et al., 2022) is widely as backbone as its significant advancement in image synthesizing. Typically, based on a pre-trained autoencoder $\mathcal{E}(\cdot)$ and $\mathcal{D}(\cdot)$, video sequences are projected into the latent space, in which a denoising network $\epsilon_\theta$ is encouraged to learn the mapping from noised video latent $z_t$ to pure video latent $z_0$. Mathematically, the noised video latent $z_t$ obeys the following distribution:

$$z_t = \sqrt{\bar{\alpha}_t} z_0 + \sqrt{1 - \bar{\alpha}_t} \epsilon, \tag{1}$$

where $\bar{\alpha}_t$ is a pre-defined parameter representing noise schedule (Ho et al., 2020), $\epsilon \sim \mathcal{N}(0, 1)$ is the added noise, and $t \sim \mathcal{U}(1, T)$ denotes time step. To restore $z_0$ from $z_t$, denoising network $\epsilon_\theta$ is forced to estimate the noise component in $z_t$, which can be expressed as:

$$\mathcal{L}(\theta) = \mathbb{E}_{z_0, \epsilon, t} \left[ \| \epsilon_t - \epsilon_\theta(z_t, c, t) \|_2^2 \right], \tag{2}$$

where $c$ represents the textual prompt of $z_0$. During sampling, $z_t$ is initialized with Gaussian noise and undergoes iterative denoising conditioned on $c$ for prompt-aligned generation.

### 3.2 TEMPORAL ATTENTION MECHANISM

The biggest difference between video generation and image generation lies in the synthesis of motion, i.e., the modeling of correlation between video sequences. This is typically achieved by temporal attention mechanism, which establishes feature interactions across frames via self-attention operations in temporal dimension. For 5D video diffusion feature $f \in \mathbb{R}^{B \times C \times F \times H \times W}$, where $B$ and $F$ represent batch axis and frame time axis, $H$ and $W$ denotes spatial resolution, temporal attention performs self-attention in its 3D reshaped variant $f' \in \mathbb{R}^{(B \times H \times W) \times C \times F}$, in which the generated attention map $\mathcal{A} \in \mathbb{R}^{(B \times H \times W) \times F \times F}$ reflects the temporal correlation between frames.

## 4 METHOD

### 4.1 TEMPORAL SELF-GUIDANCE

Temporal attention modules are extensively integrated at various hierarchical stages within the upsampling blocks of T2V architectures (Blattmann et al., 2023b; Guo et al., 2023b; Chen et al., 2023; 2024). These modules, derived from different tiers of the diffusion UNet, are employed to capture inter-frame dependencies at multiple resolutions. Although the multi-level progressive refinement approach in modeling frame-wise correlations offers advantages, our observations indicate that the temporal attention maps across different hierarchical levels can exhibit considerable discrepancies, potentially leading to structurally implausible or temporally inconsistent video outputs. To substantiate this hypothesis, we analyzed 100 structurally and motion-degraded videos alongside 100 well-generated videos. We computed the mean and standard deviation of the mean squared error (MSE) distances between the temporal attention maps of `up_blocks.1` and subsequent blocks as illustrated in Fig. 3 (a). Our findings reveal that significant disparities between temporal attention maps across different blocks are associated with the occurrence of implausible structures and temporal inconsistencies in the generated videos.

To mitigate the excessive divergence between temporal attention maps across various upsampling blocks, we introduce a straightforward yet potent temporal self-guidance mechanism. This mechanism involves the infusion of the temporal attention map of `up_blocks.1` into subsequent blocks,

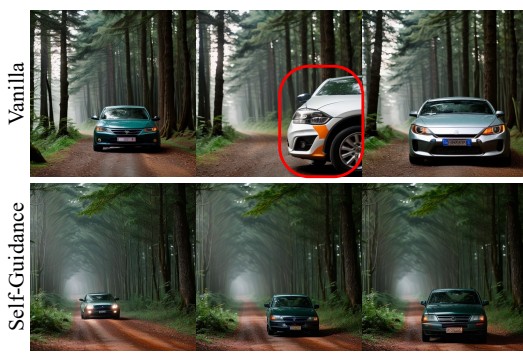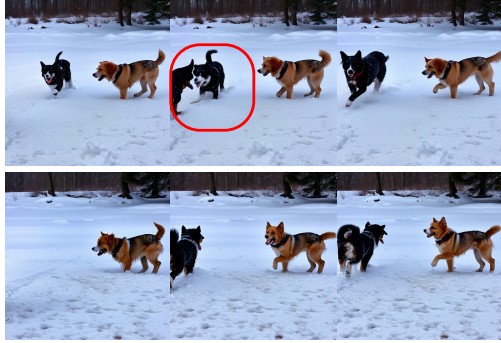

| *Car runs in the forest.* | *Two dogs are playing in the snow.* |

Figure 4: **Temporal Self-Guidance.** Temporal Self-Guidance contributes to the restoration of collapsed structures and consistency of motion in the synthesized video.

modulated by a guidance ratio $\alpha$. The adjustment is mathematically represented as:

$$\mathcal{A}_m = \mathcal{A}_m + \alpha(\mathcal{A}_1^m - \mathcal{A}_m), \tag{3}$$

where $\mathcal{A}_m$ denotes the temporal attention map of the $m$-th upsampling block ($m = 2, 3$), and $\mathcal{A}_1^m$ refers to the temporal attention map of `up_blocks.1`, which is upsampled to match the spatial dimensions of $\mathcal{A}_m$. As depicted in Fig. 3 (b) and Fig. 4, the implementation of temporal self-guidance effectively alleviates the excessive modeling disparity between different hierarchical levels of temporal attention modules, thereby diminishing structurally implausible and temporally inconsistent artifacts in the resultant video generation.

Beyond addressing the structural implausibility and temporal inconsistency issues resolved by Temporal Self-Guidance, we have observed that some generated videos, including those corrected by Temporal Self-Guidance, still suffer from a paucity of motion, often appearing nearly static. To tackle this, we introduce a novel strategy aimed at amplifying the motion amplitude and diversity within the generated videos by capitalizing on the energy inherent in the temporal attention maps.

### 4.2 FOURIER-BASED MOTION ENHANCEMENT

#### 4.2.1 ENERGY REPRESENTATION OF MOTION MAGNITUDE

The temporal attention map encapsulates a rich set of motion-related information that is pivotal for the generation of dynamic video content. We find that the energy encapsulated within the temporal attention map is indicative of the motion amplitude present in the generated video. To elaborate, consider a temporal attention map $\mathcal{A} \in \mathbb{R}^{(B \times H \times W) \times F \times F}$, where $B$ represents the batch size, $H \times W$ denotes the spatial resolution, and $F$ is the number of frames. The energy $E$ of this map can be quantified by the following equation:

$$E = \frac{1}{F} \sum_{i=0}^{F-1} \sum_{j=0}^{F-1} ||\mathcal{A}_{...,i,j}||^2, \tag{4}$$

as illustrated in Fig. 5 (a). To substantiate the correlation between the energy of the temporal attention map and the motion magnitude in the generated video, we employ the RAFT (Teed & Deng, 2020) to extract the optical flow, using the average magnitude of this flow as a metric for motion strength. Our findings reveal a positive correlation: videos with greater motion magnitudes are associated with higher energies within their temporal attention maps. This insight motivates us to manipulate the motion magnitude in the generated videos by modulating the energy intensity of the temporal attention maps. By doing so, we aim to enhance the dynamism and variability of the motion in the videos.

#### 4.2.2 MOTION ENHANCEMENT BY FREQUENCY SPECTRUM RE-WEIGHTING

To enhance the motion amplitude in generated videos by amplifying the energy of the temporal attention map, we must overcome the challenge posed by the softmax normalization inherent in attention

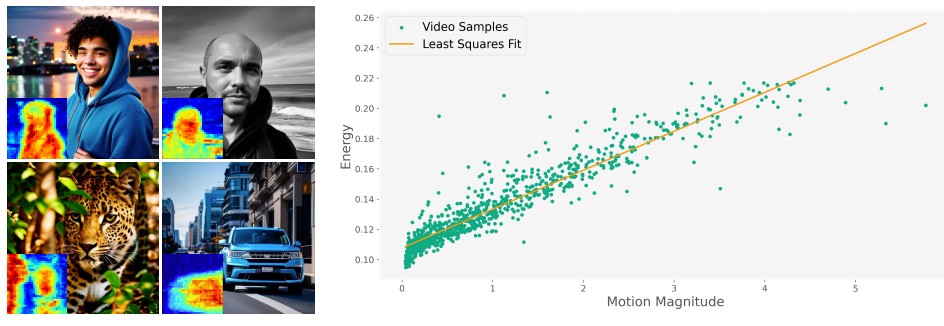

(a) Energy map of generated videos      (b) Relationship between energy and motion magnitude

Figure 5: **Visualization of energy in temporal attention map.** (a) The energy map of the generated video. (b) Videos with larger motion magnitude typically exhibit higher energy, where the motion magnitude is calculated using the optical flow of the generated videos.

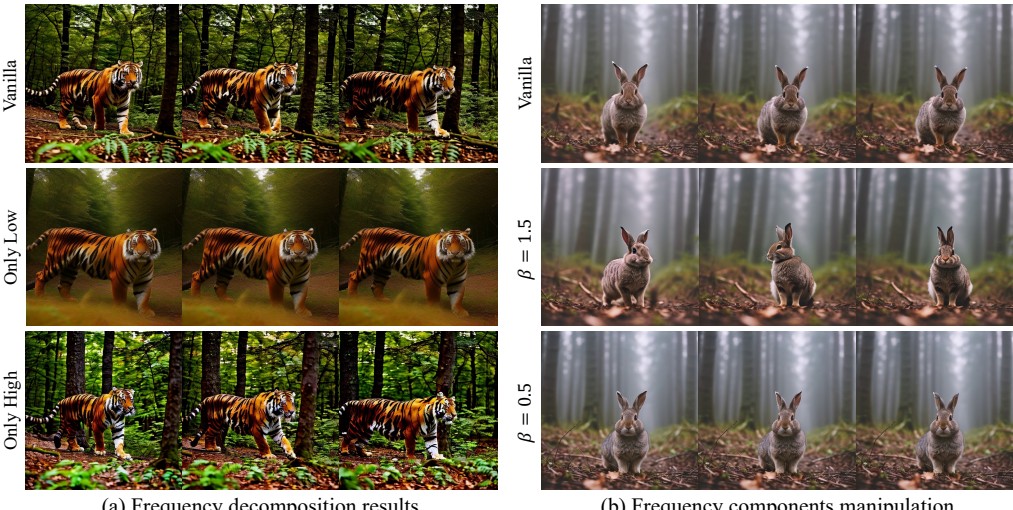

(a) Frequency decomposition results      (b) Frequency components manipulation

Figure 6: **Frequency decomposition and manipulation.** (a) Results obtained by directly removing either the high-frequency or low-frequency components from the temporal attention map. The motion in generated videos is primarily present in the high-frequency components of the temporal attention map. (b) Results obtained by scaling the high-frequency components by a factor of $\beta$.

maps, which precludes straightforward numerical scaling. To address this, we employ a sequence-to-sequence discrete frequency decomposition technique, specifically the Fast Fourier Transform (FFT), to the temporal attention map. For a given temporal attention map $\mathcal{A} \in \mathbb{R}^{(B \times H \times W) \times F \times F}$, we decompose it into its high-frequency and low-frequency components as follows:

$$
\begin{aligned}
\mathbf{A} &= \mathcal{F}(\mathcal{A}), \\
\mathbf{A}_H &= \mathbf{A}_{...,i_H}, \ i_H \in [\frac{F}{2} - \tau, \frac{F}{2} + \tau], \\
\mathbf{A}_L &= \mathbf{A}_{...,i_L}, \ i_L \in [0, \frac{F}{2} - \tau) \cup (\frac{F}{2} + \tau, F - 1],
\end{aligned}
\tag{5}
$$

where $\mathcal{F}$ denotes the FFT operation, $\mathbf{A} \in \mathbb{C}^{(B \times H \times W) \times F \times F}$ is the complex-valued matrix resulting from applying the FFT to $\mathcal{A}$, and $\tau$ is a hyperparameter that determines the frequency range for the high-pass and low-pass filters. As demonstrated in Fig. 6 (a), experiments involving the selective removal of high-frequency or low-frequency components from the temporal attention map during the denoising process have yielded insightful observations. Videos that retain only the low-frequency components tend to exhibit a nearly static structure, closely mirroring the characteristics of their unmodified counterparts. In contrast, videos that include solely high-frequency components display abundant motion but are marred by inconsistency and persistent flickering. These findings suggest that the essence of motion in generated videos is predominantly encapsulated within the high-frequency components of their temporal attention maps.

Figure 7: **BroadWay Operations. (a) Temporal Self-Guidance**. The temporal attention map from `up_blocks.1` is injected into the corresponding modules of `up_blocks.2/3` with a guidance ratio $\alpha$, in order to enhance the structural plausibility and temporal consistency. **(b) Fourier-based Motion Enhancement**. A scaling factor $\beta$ is applied to the high-frequency components of the temporal attention map, thereby amplifying the magnitude of the motion.

Motivated by these insights, we introduce a scaling factor $\beta$ to modulate the high-frequency components $\mathbf{A}_H$. The process of scaling and reconstructing the temporal attention map is formalized by the following equation:

$$\mathcal{A}^{'} = \widetilde{\mathcal{F}}(\beta \mathbf{A}_H + \mathbf{A}_L), \tag{6}$$

where $\widetilde{\mathcal{F}}$ represents the inverse Fast Fourier Transform (iFFT) operation, and $\mathcal{A}^{'}$ signifies the temporal attention map with the scaled high-frequency components. Based on aforementioned equations, the following theorems can be proven. Please refer to Section A.1.1 for a comprehensive proof.

**Theorem 1.** *For any $\beta \geq 0$, $\mathcal{A}^{'}$ possesses the softmax property. Specifically, $\sum_k \mathcal{A}^{'} = \sum_k \mathcal{A} = \mathbf{I}$, where $k$ denotes the softmax dimension associated with $\mathcal{A}$, and $\mathbf{I}$ is an all-ones matrix.*

Therefore, $\mathcal{A}^{'}$ can replace $\mathcal{A}$ as the new temporal attention map in the decoder.

**Theorem 2.** *If $\beta > 1$, then the energy of $\mathcal{A}^{'}$, denoted as $E_x^{'}$, is greater than the energy of $\mathcal{A}$, denoted as $E_x$. Conversely, if $0 < \beta < 1$, then $E_x^{'}$ is less than $E_x$.*

As illustrated in Fig. 6 (b), setting $\beta = 1.5$ amplifies the energy of the temporal attention maps, leading to greater motion magnitude. Conversely, setting $\beta = 0.5$ results in reduced motion.

## 4.3 BROADWAY

Leveraging Temporal Self-Guidance and Fourier-based Motion Enhancement, we introduce Broad-Way, a parameter-free method that enhances the quality of text-to-video generation without increasing memory requirements or sampling time. As illustrated in Fig. 7, BroadWay initially applies Temporal Self-Guidance to improve the structural coherence and temporal consistency of the video. Subsequently, Fourier-based Motion Enhancement is employed to amplify motion dynamics. To ensure that the motion magnitude of generated videos processed by BroadWay exceeds that of the original, unenhanced videos, the energy of the temporal attention map after Fourier-based Motion Enhancement, denoted as $E_3$, must be greater than the energy of the original temporal attention map, denoted as $E_1$. To achieve this, the scaling factor $\beta$ is defined as a function of the energies before and after Temporal Self-Guidance, $E_1$ and $E_2$, respectively:

$$\beta(E_1, E_2) = max\{\beta_0, \sqrt{\frac{E_1 - E_2^L}{E_2^H}}\}, \tag{7}$$

where $\beta_0$ is user-given value of $\beta$ to control the motion magnitude. $E_2^H$ and $E_2^L$ denoting the energies of the high-frequency and low-frequency of the attention map after applying Temporal Self-Guidance, respectively. Please refer to Section A.1.2 for a detailed proof for Eq. 7.

## 5 EXPERIMENTS

### 5.1 EXPERIMENTS SETUP

**Setting up.** We mainly conduct our experiments on two mainstream diffusion based T2V backbones with superior visual quality: AnimateDiff ($512 \times 512$) (Guo et al., 2023b) and VideoCrafter2 ($320 \times$

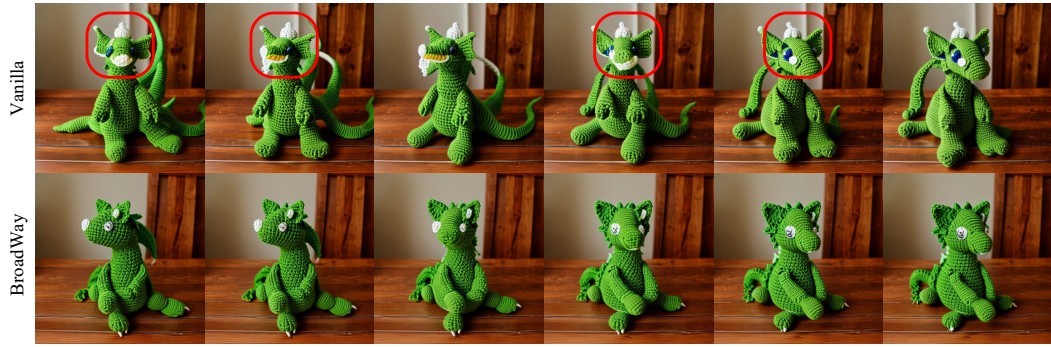

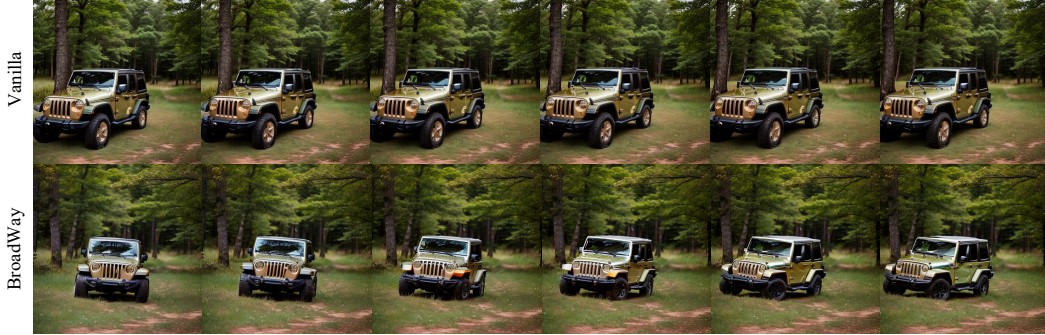

*A jeep driving on the grass near a forest ...*

Figure 8: **Samples synthesized by AnimateDiff with or without BroadWay.** The samples utilizing the BroadWay exhibit enhanced structural plausibility, temporal consistency, and an increased richness in motion dynamics.

512) (Chen et al., 2024). Results synthesized by vanilla T2V backbones are used as a benchmark. The BroadWay parameters are set to $\alpha = 0.6$, $\beta = 1.5$ in default for AnimateDiff, $\alpha = 0.1$, $\beta = 10$ in default for VideoCrafter2. BroadWay operations are only applied during the first 20% steps of the denoising process. DDIM sampler (Song et al., 2020) with classifier-free guidance (Ho & Salimans, 2022) is adopted in the inference phase.

**Evaluation metrics.** We report three metrics for quantitative evaluation. First, we conduct a user study with 30 participants to assess *Video Quality*, considering both structure coherence and motion magnitude. Secondly, we compare the *Optical Flow* values of 1000 videos generated by Vanilla T2V backbones and BroadWay-enhanced backbones. Additionally, we employ a multimodal large language model, GPT-4o (Achiam et al., 2023), for a comprehensive Multimodal-Large-Language-Model (MLLM) Assessment on hundreds of generated videos. Refer to Section A.3 for details.

## 5.2 Qualitative Comparison

As presented in Fig. 8 and Fig. 9, with the integration of BroadWay, various T2V backbones demonstrates a notable performance improvement compared to their vanilla synthesis results. For instance, giving AnimateDiff the prompt "*a green wool doll is displayed on the wooden table.*", BroadWay enhances the structural consistency of the synthesized video, preventing the collapse of the doll's head and tail. Moreover, in the "*A jeep driving on the grass near a forest.*" case, BroadWay amplifies the dynamic effects of the scene, making the jeep exhibit more pronounced motion. For VideoCrafter2, when provided with the prompt "*A horse jumping over a fence during a race, crowd cheering.*", BroadWay reconstructs the structure of the rider and horse, addressing the issue of structural anomalies in the horse's legs while enhancing the overall motion to appear more synchronized and aesthetically pleasing. In cases like "*A penguin sliding on ice, snowy landscape in the background.*", BroadWay preserves the original structural integrity while introducing richer, more dynamic motion to the scene.

In summary, BroadWay effectively improves the structural consistency of synthesized videos while amplifying their motion dynamics, resulting in a significant enhancement in the overall synthesis quality of the T2V backbones.

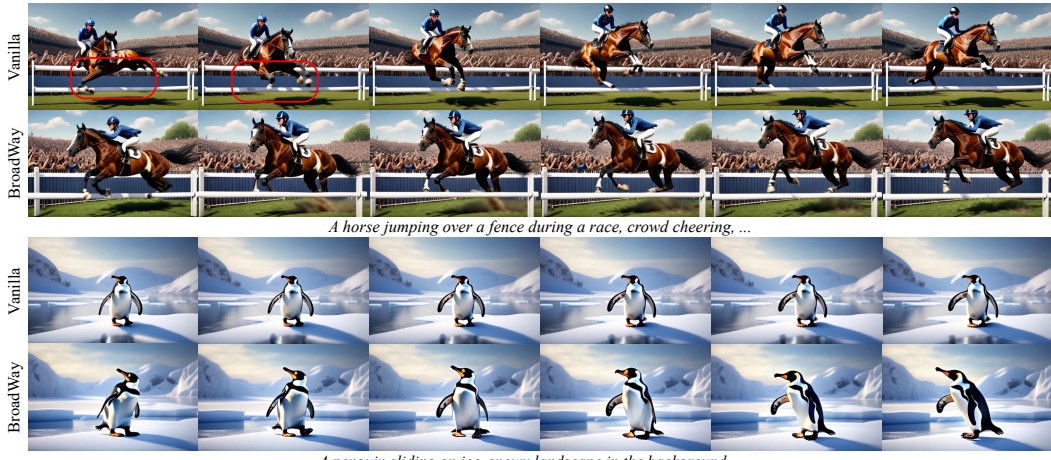

Figure 9: **Samples synthesized by VideoCrafter2 with or without BroadWay.** The samples utilizing the BroadWay exhibit enhanced structural plausibility, temporal consistency, and an increased richness in motion dynamics.

## 5.3 QUANTITATIVE EVALUATION

**User Study.** As shown in Table 1 (a), we present the voting results, expressed as percentages, for vanilla T2V backbones and BroadWay-enhanced backbones. Our analysis shows that BroadWay receives the majority of votes, demonstrating that BroadWay provides a substantial improvement to the T2V diffusion model in terms of overall video quality, taking into account both structure coherence and motion magnitude.

**Motion Magnitude.** To objectively evaluate the motion magnitude, RAFT (Teed & Deng, 2020) is introduced to estimate the forward optical flow between consecutive frames, and the average intensity value of estimated optical flow is used to quantify the motion magnitude within the video. As presented in Table 1 (b) BroadWay shows substantial improvements in mean motion intensity, indicates its efficacy in producing large-magnitude motion.

**MLLM Assessment.** In light of the impressive strides made by Multimodal-Large-Language-Models (MLLM) recently in image/video understanding, the state-of-the-art MLLM, i.e., GPT-4o (Achiam et al., 2023), is employed for video quality assessment, covering structural rationality and motion consistency. As can be observed in Table 1 (c)-(d), BroadWay exhibits notable gains in both metrics, validating its role in substantially improving overall video quality.

Table 1: Quantitative results with or without BroadWay.

| Method | (a) Video Quality | (b) Optical Flow | (c) Structural Rationality | (d) Motion Consistency |
|---|---|---|---|---|
| AnimateDiff | 25.42% | 1.5743 | 41.94% | 34.62% |
| **+ BroadWay** | **74.58%** | **2.4673** | **58.06%** | **65.38%** |
| VideoCrafter2 | 30.54% | 1.5555 | 18.48% | 39.60% |
| **+ BroadWay** | **69.46%** | **3.6204** | **81.52%** | **60.40%** |

## 5.4 ABLATION STUDY

**Effects of Temporal Self-Guidance (TSG).** Temporal Self-Guidance plays a critical role in reinforcing structural integrity, thus effectively mitigating structural breakdowns and preventing motion artifacts, as can be observed in Fig. 10 (a). However, it cannot enhance the magnitude of motion, showing limited improvement in scenarios with little motion, as shown in Fig. 10 (b).

**Effects of Fourier-based Motion Enhancement (FME).** Fourier-based Motion Enhancement is responsible for amplifying the motion dynamics in generated videos. In scenarios where the motion is insufficient, this technique effectively increases the dynamic content, as shown in Fig. 10 (b). However, motion enhancement alone does not guarantee appealing video quality when structural breakdown occurs, as illustrated in Fig. 10 (a).

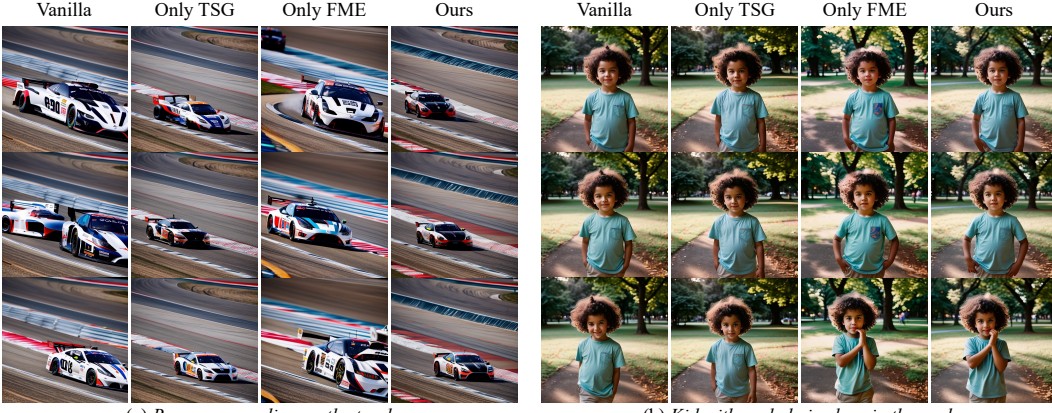

(a) *Race car speeding on the track.*   (b) *Kid with curly hair plays in the park.*

Figure 10: **Ablation study on BroadWay components.** The left panel illustrates an instance of inconsistency artifacts present in the original video, whereas the right panel exhibits a scenario where the original video lacks sufficient motion.

**Effects of BroadWay.** By integrating Temporal Self-Guidance with Fourier-based Motion Enhancement, BroadWay achieves simultaneous enhancement of both the structural integrity and motion dynamics in generated videos (Fig. 10 (a)-(b) Ours *vs.* Vanilla).

## 5.5 IMAGE-TO-VIDEO

Similar to text-to-video (T2V) tasks, image-to-video (I2V) is also a significant research area within video diffusion models. Here we employ SparseCtrl (Guo et al., 2023a), a strong and flexible structure control method, as the I2V backbone to preliminarily validate the potential of BroadWay in image-to-video tasks. As illustrated in Fig. 11, the infusion of BroadWay into SparseCtrl serves to enhance the dynamic effects of the synthesized video while preserving the structural integrity of the reference image. Specifically, we observe that the video synthesized with BroadWay exhibits more vivid wave motions, and the reflections of the setting sun display enhanced dynamic aesthetics. These experimental results demonstrate that BroadWay effectively enhances the quality of both T2V and I2V video generation tasks, positioning it as a versatile and powerful booster for video diffusion models.

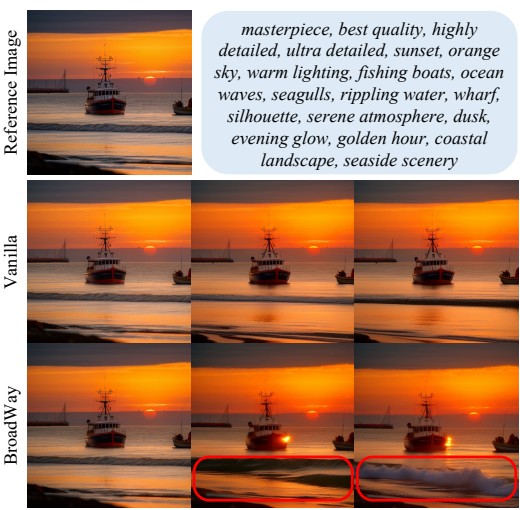

*masterpiece, best quality, highly detailed, ultra detailed, sunset, orange sky, warm lighting, fishing boats, ocean waves, seagulls, rippling water, wharf, silhouette, serene atmosphere, dusk, evening glow, golden hour, coastal landscape, seaside scenery*

Figure 11: Generated results by SparseCtrl with or without BroadWay.

## 6 CONCLUSION

In this study, we present BroadWay, a training-free method to improve the quality of text-to-video generation without introducing additional parameters, augmenting memory or sampling time. BroadWay is composed of Temporal Self-Guidance and Fourier-based Motion Enhancement. The former improves the structural plausibility and temporal consistency by reducing the disparity between the temporal attention maps across various decoder blocks. The later enhances the magnitude and richness of motion by scaling the high frequency of the temporal attention maps. The proposed method can be easily integrated with other T2V models in a plug-and-play manner, offering a general and effective solution to enhance video generation quality during inference phase.

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

# A APPENDIX

In the appendix, we present the proof of Fourier-based Motion Enhancement (Section A.1), additional qualitative results (Section A.2), details of our quantitative evaluation (Section A.3), as well as the limitations of our method(Section A.4), as a supplement to the main paper.

## A.1 PROOF OF FOURIER-BASED MOTION ENHANCEMENT

In this section, we provide a detailed proof of how Fourier-based Motion Enhancement alters the energy of the temporal attention map in the denoising process.

### A.1.1 FREQUENCY MANIPULATION

Given a temporal attention map $\mathcal{A} \in \mathbb{R}^{(B \times H \times W) \times F \times F}$ with batch size $B$, spatial resolution $H \times W$ and frame number $F$, since we treat it as a batch of attention sequences, we will next discuss the operations performed on a single `softmax` sequence $x[n]$ of length $F$.

Mathematically, the operation of mapping the sequence $x[n]$ to the frequency domain is performed by the Discrete Fourier Transform (DFT):

$$X[k] = \sum_{n=0}^{F-1} x[n] \cdot e^{-j \frac{2\pi}{N} kn}, \ k = 0, 1, \ldots, F-1, \tag{8}$$

Parseval's theorem states that the energy of a sequence is preserved under frequency domain transformation, meaning that the energy $E_x$ of sequence $x[n]$ is the same in both the time and frequency domains. This theorem can be expressed as follows:

$$E_x = \sum_{n=0}^{F-1} x[n]^2 = \frac{1}{F} \sum_{k=0}^{F-1} X[k]^2, \tag{9}$$

As mentioned in Section 4.2.2, Fourier-based Motion Enhancement uses a threshold index $\tau$ to separate the high-frequency and low-frequency components of the sequence, scaling the high-frequency components by a factor of $\beta$. This operation can be expressed as:

$$X'[k] = \begin{cases} \beta \cdot X[k] & k \in [\frac{F}{2} - \tau, \frac{F}{2} + \tau], \\ X[k] & otherwise, \end{cases} \tag{10}$$

After applying this manipulation, the energy $E_x'$ of current attention sequence $x'[n]$ is given by:

$$E_x' = \frac{1}{F} [ \sum_{k \notin [\frac{F}{2} - \tau, \frac{F}{2} + \tau]} X^2[k] + \beta^2 \sum_{k \in [\frac{F}{2} - \tau, \frac{F}{2} + \tau]} X^2[k]], \tag{11}$$

Thus the energy change amount $\Delta E$ caused by Fourier-based Motion Enhancement can be computed as:

$$\Delta E = E_x' - E_x$$
$$= \frac{(\beta^2 - 1)}{F} \sum_{k \in [\frac{F}{2} - \tau, \frac{F}{2} + \tau]} X^2[k],$$

Clearly, in the scenario where $\beta > 1$, Fourier-based Motion Enhancement will lead to an increase in the energy of the attention sequence ($\Delta E > 0$), while the opposite will result in a decrease in energy ($\Delta E < 0$), which elucidates the mechanism by which Fourier-based Motion Enhancement effectively enhances motion magnitude in synthesized videos.

Furthermore, it can be demonstrated that the attention sequence processed by Fourier-based Motion Enhancement remains a `softmax` sequence. This property is preserved because the DC component $X[0]$ of the attention sequence, which determines the sum of the sequence, is not modified throughout the operation. By plugging $k = 0$ into Eq. 8, we can ascertain this property:

$$X[0] = \sum_{n=0}^{F-1} x[n] = \sum_{n=0}^{F-1} x'[n] = 1, \tag{12}$$

A.1.2 ADAPTIVE $\beta$

As depicted in Fig. 7, let $E_1$ denote the the energy of the temporal attention map before applying BroadWay operations, $E_2$ the energy after Temporal Self-Guidance, and $E_3$ the energy after Fourier-based Motion Enhancement. Here, we demonstrate that using the adaptive $\beta$ as defined in Eq. 7 ensures that $E_3 \geq E_1$.

Based on the separation of high-frequency and low-frequency components in the sequence as described in Section A.1.1, we can compute the energy of the high-frequency and low-frequency parts of the sequence $x[n]$, denoted as $E_x^H$ and $E_x^L$, respectively:

$$
\begin{aligned}
E_x^H &= \frac{1}{F} \sum_{k \in [\frac{F}{2} - \tau, \frac{F}{2} + \tau]} X^2[k], \\
E_x^H &= \frac{1}{F} \sum_{k \notin [\frac{F}{2} - \tau, \frac{F}{2} + \tau]} X^2[k],
\end{aligned}
\tag{13}
$$

According to Eq. 9 and Eq. 13, it is evident that the following relationship holds:

$$
E_x = E_x^H + E_x^L,
\tag{14}
$$

Furthermore, we can concisely express the energy manipulation performed by Fourier-based Motion Enhancement described in Section A.1.1, as follows:

$$
E_x^{'} = \beta^2 E_x^H + E_x^L,
\tag{15}
$$

which indicates:

$$
E_3 = \beta^2 E_2^H + E_2^L,
\tag{16}
$$

Therefore, to ensure $E_3 \geq E_1$, it is necessary to ensure that $\beta$ adheres to the following condition:

$$
\beta^2 E_2^H + E_2^L \geq E_1,
\tag{17}
$$

The critical value of $\beta$, denoted as $\beta_c$, that satisfies this condition is:

$$
\beta_c = \sqrt{\frac{E_1 - E_2^L}{E_2^H}},
\tag{18}
$$

In BroadWay operations, the user-specified $\beta$, denoted as $\beta_0$, will be compared with the critical value $\beta_c$, and the larger of the two will be selected as the actual $\beta$ value in Fourier-based Motion Enhancement:

$$
\beta = \begin{cases} \beta_0 & \beta_0 \geq \beta_c, \\ \beta_c & \beta_0 < \beta_c, \end{cases}
\tag{19}
$$

By adopting such a adaptive $\beta$ value, it can be theoretically guaranteed that the energy of the temporal attention map is increased during BroadWay operations, thereby enhancing the motion magnitude in synthesized videos.

## A.2 ADDITIONAL QUALITATIVE RESULTS

In this section, we provide additional qualitative comparison results of BroadWay on AnimateDiff (Fig. 12, Fig. 13, Fig. 17 and Fig. 14) and VideoCrafter2 (Fig. 15, Fig. 16, Fig. 18).

## A.3 MATERIALS USED IN QUANTITATIVE EXPERIMENTS

**User Study Details.** In our user study, each participant receives 50 videos synthesized by Vanilla T2V backbones and 50 videos synthesized by BroadWay-enhanced backbones. These videos are sampled from the same random seeds to ensure fair comparison. For each video pair from Vanilla and Vanilla+BroadWay, participants are required to select the video they perceive as superior based on overall *Video Quality*, considering both structure coherence and motion magnitude, and cast their vote accordingly. The videos were presented in a randomized order to reduce potential bias, and participants were allowed ample time to review each pair before making their selections.

**MLLM Prompt.** Here, we present the prompt used in the MLLM assessment.

```
query = """
You are provided with two sets of video frames, each containing
4 representative frames, along with a shared textual prompt that
was used to generate both videos.
Your task is to perform a comparative evaluation of the two videos,
focusing on their structure rationality / motion consistency.

""".strip()

prefix_1 = """
Here is the frame data of video_1:

"""

prefix_2 = """
Here is the frame data of video_2:

"""

suffix = """
Based on your evaluation of motion consistency, choose the video
set you find to be superior.
If you determine that the first set of frames (Video_1) is better,
respond with "A". If the second set (Video_2) is superior, respond
with "B". Return only "A" or "B" based on your assessment.
"""
```

## A.4 LIMITATIONS

**Parameter Sensitivity.** The default values of BroadWay parameters $\alpha$ and $\beta$ are relatively robust within a specific T2V backbone but may not be universally optimal for different backbones. Users seeking enhanced visual quality are encouraged to manually adjust these parameters. Increasing $\alpha$ can lead to stronger motion dynamics, while a higher value of $\beta$ enhances structural consistency.

**Performance Upper Bound.** Although BroadWay demonstrates the capability to unlock the synthesis potential of various T2V backbones, the synthesized videos remain confined within the sampling distribution of the original T2V backbone. Therefore, the upper performance bound of our proposed method is still constrained by the original T2V backbone.

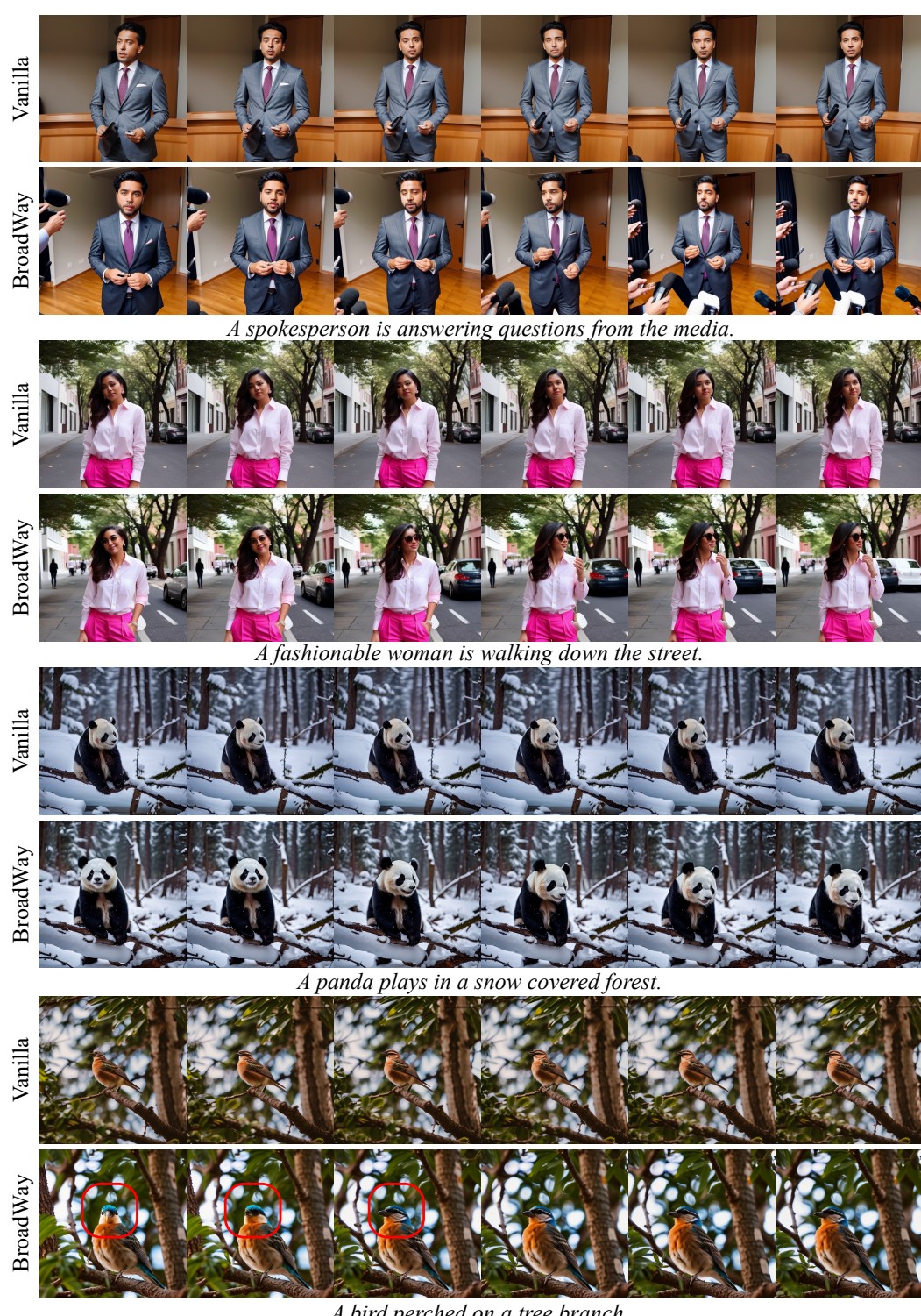

*A spokesperson is answering questions from the media.*

*A fashionable woman is walking down the street.*

*A panda plays in a snow covered forest.*

*A bird perched on a tree branch.*

Figure 12: **More results on AnimateDiff (Object Motion Enhancement). Please refer to the supplementary materials for best view.**

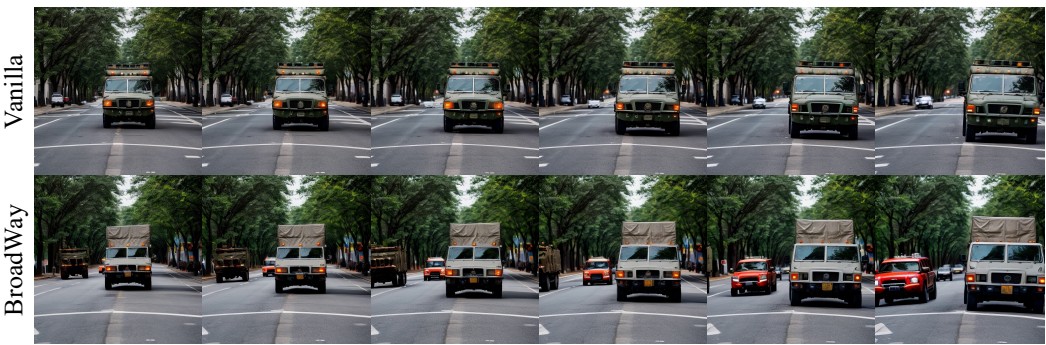

*A military truck is driving down the street.*

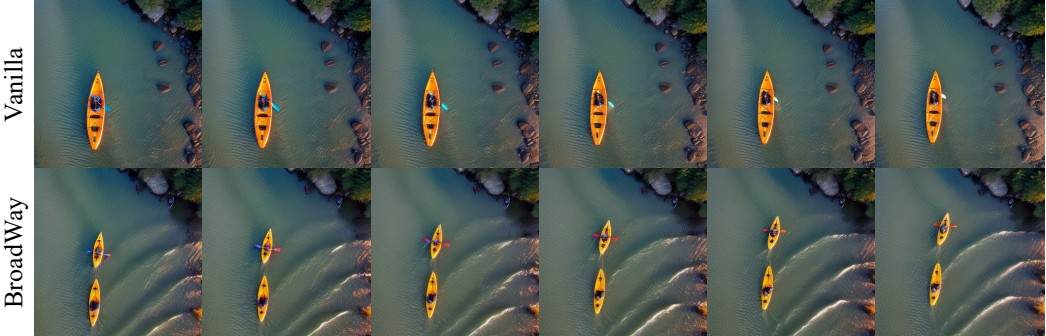

*Kayakers paddling down a river.*

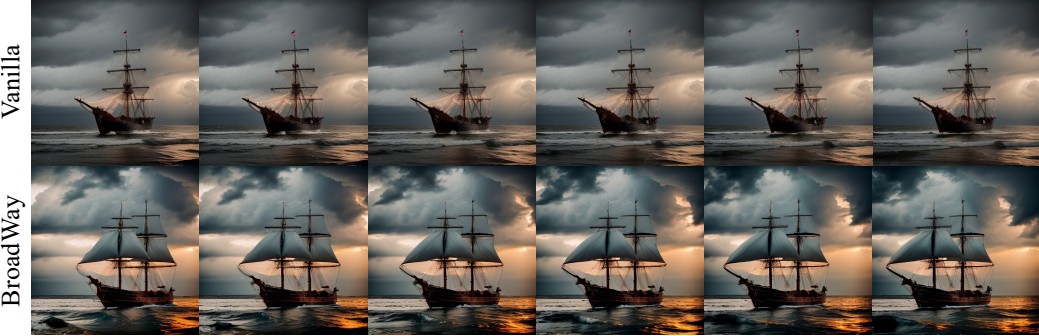

*An excavator is digging at a construction site.*

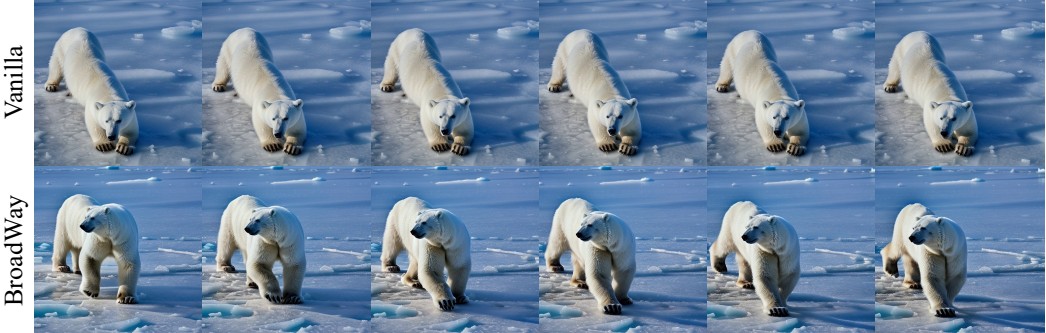

*A polar bear walks on the ice surface.*

Figure 13: **More results on AnimateDiff (Object Motion Enhancement). Please refer to the supplementary materials for best view.**

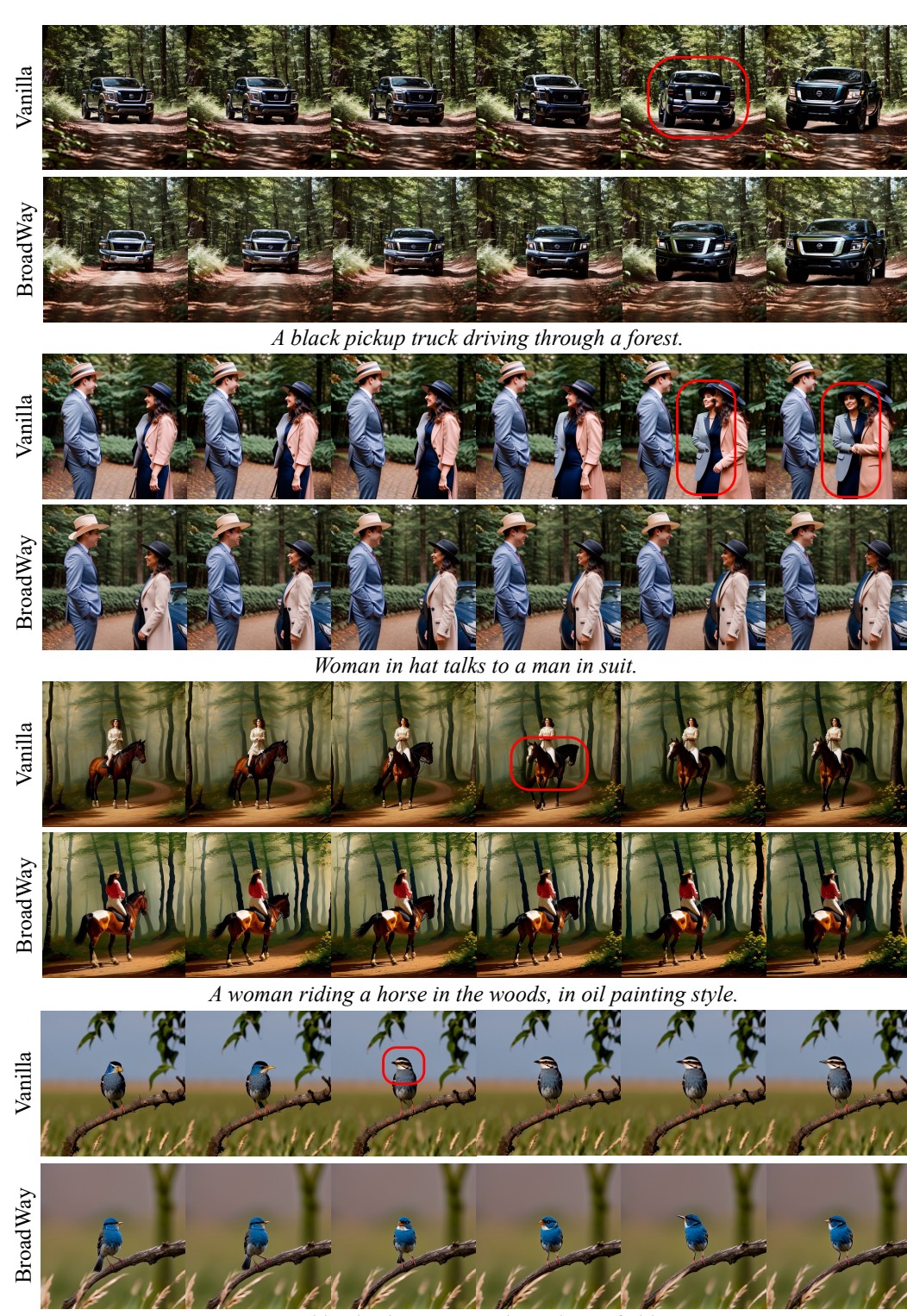

Figure 14: **More results on AnimateDiff (Corrupted Case Repair). Please refer to the supplementary materials for best view.**

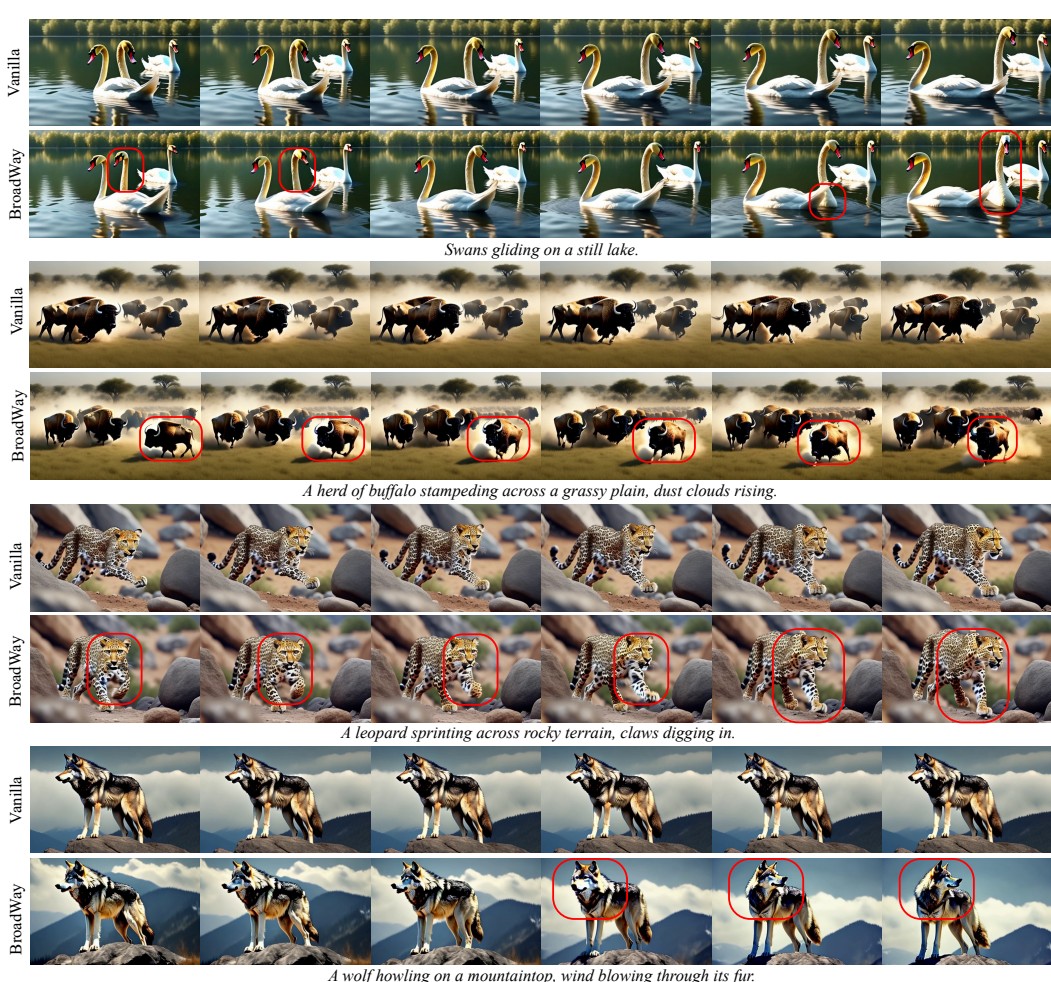

Figure 15: **More results on VideoCrafter2 (Object Motion Enhancement). Please refer to the supplementary materials for best view.**

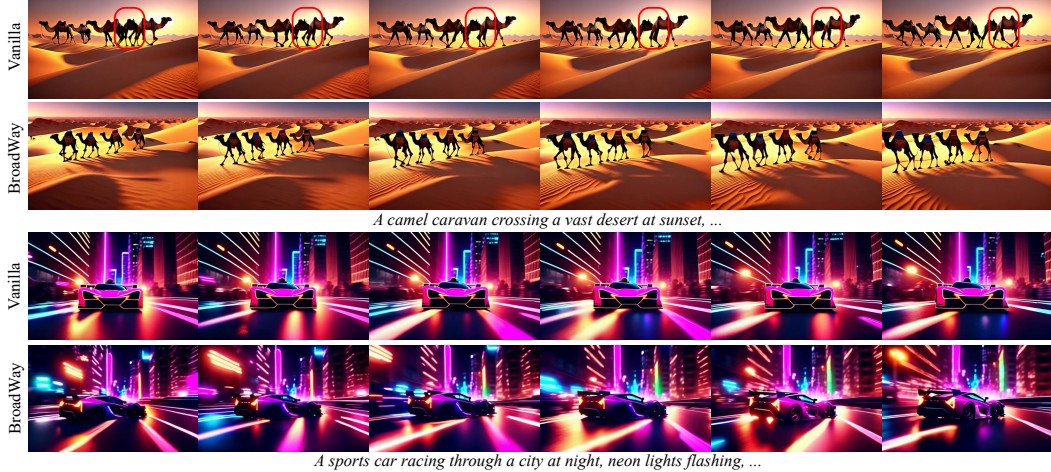

Figure 16: **More results on VideoCrafter2 (Corrupted Case Repair). Please refer to the supplementary materials for best view.**

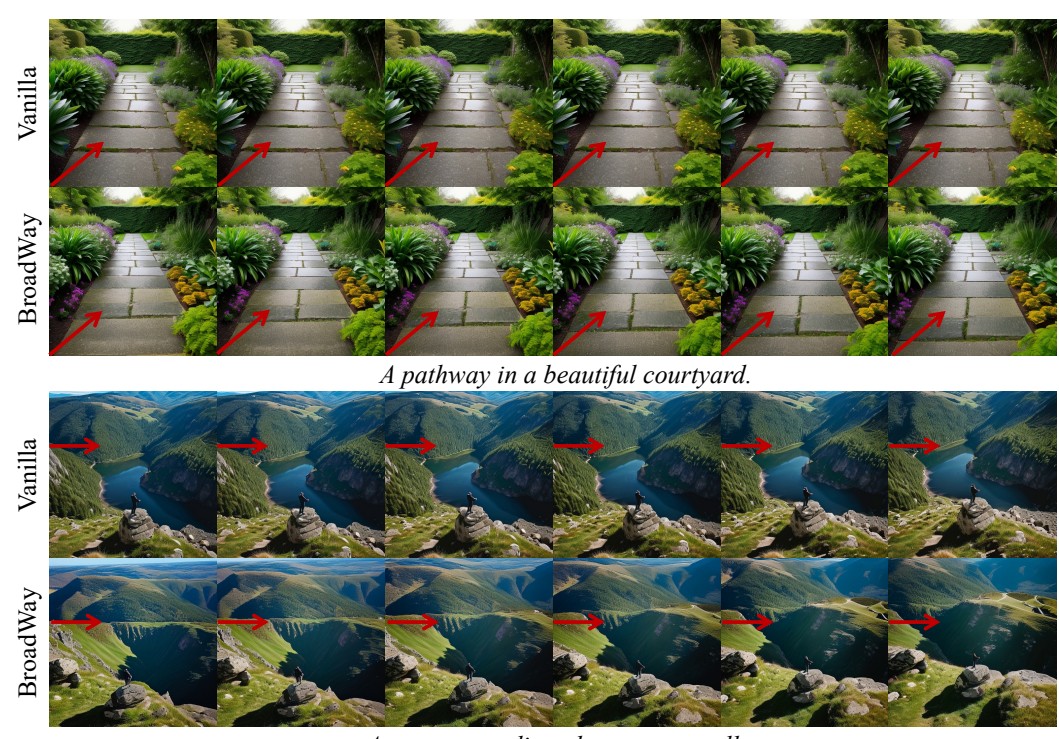

*A pathway in a beautiful courtyard.*

*A person standing above a vast valley.*

Figure 17: **More results on AnimateDiff (Camera Motion Enhancement). Please refer to the supplementary materials for best view.**

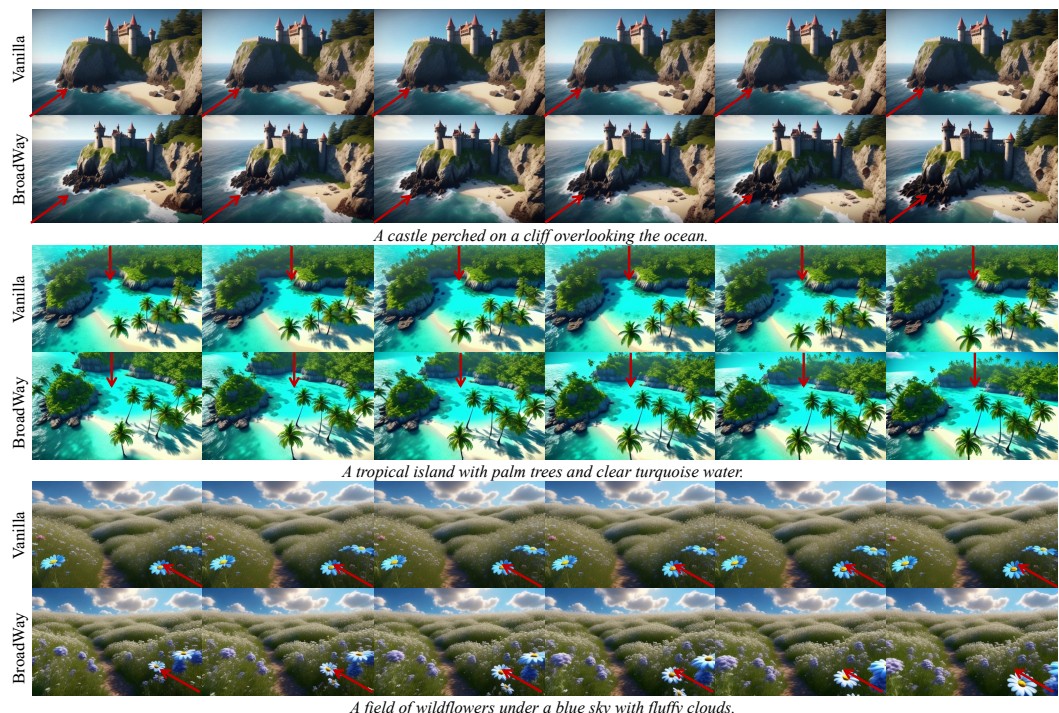

*A castle perched on a cliff overlooking the ocean.*

*A tropical island with palm trees and clear turquoise water.*

*A field of wildflowers under a blue sky with fluffy clouds.*

Figure 18: **More results on VideoCrafter2 (Camera Motion Enhancement). Please refer to the supplementary materials for best view.**

