# OpenReview forum: "BroadWay: Boost Your Text-to-Video Generation Model in a Training-free Way"
_ICLR.cc/2025/Conference — ICLR 2025 Conference Withdrawn Submission_

### Official Review · Reviewer_ZCUj · 2024-10-27

**Soundness:** 3
**Presentation:** 3
**Contribution:** 3
**Rating:** 5
**Confidence:** 5

**Summary:**

This paper introduces BroadWay, a training-free method to improve T2V video quality, especially structural plausibility, temporal and consistency. BroadWay consists of two components: Temporal Self-Guidance, which enhances structural plausibility and temporal consistency by reducing disparities in temporal attention maps, and Fourier-based Motion Enhancement, which amplifies motion by increasing the energy in the maps. Experiments and results seem promising.

**Strengths:**

1. This paper tries to solve an important question of motion quality in video generation. The authors propose a new method to modify temporal attention maps using self-guidance and frequency spectrum re-weighting. The method itself seems reasonable and interesting.
2. The results shown in the paper seem promising and superior to baseline model.
3. The paper writing is clear and very easy to follow.

**Weaknesses:**

1. My primary concern lies with the reasoning and justification for the proposed method. Many of the descriptions and claims in the paper appear to be rather ad hoc, lacking detailed analysis or theoretical proof to substantiate their validity.

2. The authors clearly described their method. However, it is not very clear for the reason and hyper-parameter choice for each step.
    I think there must be some explanation under these steps, and it would be much better if more discussion and explanations are provided.

3. The results in the main paper seem promising. However, the videos in the supplementary material seem to be slightly better, especially in consitency and motion magnitude, but still contain artifacts. I wonder maybe the proposed temporal attention map modifications are not essential to this problem?

**Questions:**

1. I am concerned about the reasoning and justification for the proposed method.
    a. In L83, the authors claim temporal inconsistency and motion artifacts are related to temporal attention map disparity. However, why such disparity lead to artifacts? In Sec 4.2, the authors also modify and amplify several frequency components, why such operation not lead to disparity and degrades results?
    b. In L87~89, the meaning of `energy` is not defined and unclear. In the second row of Fig. 2, the background region slightly translates, but shows a large response similar to foreground cat, why is that? Is this `rich motion`?
    c. In L210, why to choose up blocks.1 as anchor? Does temporal attention maps in different blocks share similar meaning and can be computed as in Equ (3)?
    d. In Sec 4.2, why the authors choose to modify in frequency space? The artifacts or motion inconsitency do not necessary happen in frequency domain.
    e. In Equ (5), A is a 3D tensor, it is not clear how to do the fourier transform. Does that mean a 1D FFT along the last axis?

---

### Official Review · Reviewer_yzcA · 2024-10-29

**Soundness:** 2
**Presentation:** 3
**Contribution:** 2
**Rating:** 3
**Confidence:** 5

**Summary:**

This paper proposes a training-free method called BroadWay to improve the quality of text-to-video (T2V) generation models without introducing additional parameters, memory, or sampling time. BroadWay consists of two components: Temporal Self-Guidance and Fourier-based Motion Enhancement. The former improves structural plausibility and temporal consistency by reducing the disparity between temporal attention maps across decoder blocks. The latter enhances motion magnitude and richness by scaling the high-frequency components of the temporal attention maps. The authors demonstrate the effectiveness of BroadWay on various T2V backbones, including AnimateDiff and VideoCrafter2, and show that it can also be applied to image-to-video (I2V) tasks.

**Strengths:**

1. The authors demonstrate the effectiveness of BroadWay on various T2V backbones and I2V tasks, showing its versatility and potential for widespread adoption.
2. This paper is well-organized, and the authors provide a clear explanation of the proposed method, making it easy to follow.

**Weaknesses:**

1.	Firstly, this paper focuses on methods that do not require training to improve the performance of video diffusion models, which has been explored by existing works, including FreeInit [1], UniCtrl [2], and I4VGen [3]. However, these works are not adequately discussed in this paper. In fact, [1] also explores the motion degree of generated videos from the perspective of high and low frequency decoupling, and [2] studies the attention layer of video diffusion models. However, the omission of these works means that this paper cannot be considered a well-prepared version for ICLR. This also affects the evaluation of the novelty of this paper.
2.	Experiments. In fact, there are many benchmarks focused on video generation, such as T2V-CompBench [4] and VBench [5], which provide reliable evaluation metrics. However, this paper hardly provides quantitative evaluation results. For the evaluation of the motion degree of generated videos, the "Dynamic Degree" in VBench provides a reference.
3.	Ablation experiments. Providing only visualization results in Figure 10 is not convincing, and more results, including quantitative results, are necessary. Moreover, in line 215, the authors should provide more discussion on the selection of up_blocks.1.
4.	In the limitations section, the authors also mention that the proposed method is parameter-sensitive, and it would be better to provide more experimental results on this issue.
5.	I have doubts about the results in Table 2(b). For AnimateDiff, when LoRA is not introduced, the generated videos are temporally chaotic, corresponding to a higher motion degree. Therefore, FreeInit and others introduce Realistic Vision V5.1 LoRA as the video baseline, which is not mentioned in this paper. Therefore, the authors need to provide more explanations for Table 2.

[1] FreeInit: Bridging Initialization Gap in Video Diffusion Models, Wu et al., ECCV 2024
[2] UniCtrl: Improving the Spatiotemporal Consistency of Text-to-Video Diffusion Models via Training-Free Unified Attention Control, Chen et al., arXiv 2024
[3] I4VGen: Image as Free Stepping Stone for Text-to-Video Generation, Guo et al., arXiv 2024
[4] T2V-CompBench: A Comprehensive Benchmark for Compositional Text-to-video Generationrk, Sun et al., arXiv 2024
[5] VBench: Comprehensive Benchmark Suite for Video Generative Models, Huang et al., CVPR 2024

**Questions:**

In conclusion, considering the weaknesses mentioned above, this paper cannot be considered a well-prepared version for ICLR. Therefore, I lean towards rejecting this manuscript.

---

### Official Review · Reviewer_An3y · 2024-11-04

**Soundness:** 3
**Presentation:** 3
**Contribution:** 3
**Rating:** 6
**Confidence:** 4

**Summary:**

This paper presented a training-free method BroadWay to improve the quality of text-to-video generation. BroadWay involves two major components, Temporal Self-Guidance and Fourier-based Motion Enhancement. Experiments show its effectiveness on several existing video generation models such as AnimateDiff and VideoCrafter2.

**Strengths:**

+ A training-free method BroadWay to improve the quality of text-to-video generation.
+ Temporal Self-Guidance and Fourier-based Motion Enhancement.
+ Effectiveness on several existing video generation models such as AnimateDiff and VideoCrafter2.

**Weaknesses:**

- Will the method be extended to DiT-based video generation.
- Several relevant training-free enhancement methods, such as Freeu and VideoElevator, also adopted Fourier-based Enhancement. More discussion and comparison is suggested to clarify their difference.
- Albeit training-free is convenient for use, training usually is beneficial to performance. Some discussion on this issue is preferred.

**Questions:**

- Will the method be extended to DiT-based video generation.
- Several relevant training-free enhancement methods, such as Freeu and VideoElevator, also adopted Fourier-based Enhancement. More discussion and comparison is suggested to clarify their difference.
- Albeit training-free is convenient for use, training usually is beneficial to performance. Some discussion on this issue is preferred.

---

### Official Review · Reviewer_CMvV · 2024-11-04

**Soundness:** 1
**Presentation:** 3
**Contribution:** 2
**Rating:** 3
**Confidence:** 5

**Summary:**

The paper proposed a training-free method for improving both temporal consistency and motion intensity of video diffusion models. The first component is temporal self-guidance. The analysis shows that disparities between temporal attention maps across different blocks are related to the structure coherence of generated videos. To create videos with better structure coherence and temporal consistency, the temporal attention map of the first upsampling block is added to the attention maps of subsequent blocks. The second component is frequency spectrum re-weighting. The attention maps are decomposed into a high-frequency part and a low-frequency part. To increase the motion intensity, the high-frequency part will be multiplied by a scalar greater than 1.

**Strengths:**

- The analysis of the temporal attention map is interesting and provides a lot of insights.
- The paper reads well and is easy to follow.
- The ideas of both components are interesting.

**Weaknesses:**

- There are only ~10 example video results for each model in the supplemental material.
- For different models (AnimateDiff, VideoCrafter2), the default hyper-parameters are quite different. Users might need heavy manual tuning for these hyper-parameters.
- Ablation experiments of different values of alpha and beta are missing.
- VBench is a standard video generation benchmark, but the paper doesn't use the metrics from VBench.

**Questions:**

How is the hyper-parameter \tau decided? How does it affect the performance of the method?

---

### Official Review · Reviewer_tFqM · 2024-11-04

**Soundness:** 3
**Presentation:** 4
**Contribution:** 3
**Rating:** 6
**Confidence:** 5

**Summary:**

This paper proposes a training-free, plug-and-play approach called BroadWay to enhance text-to-video (T2V) generation models. The proposed method is designed to address common challenges including structural artifacts, temporal inconsistencies, and limited motion dynamics in T2V models, by manipulating temporal attention maps. The two main components are Temporal Self-Guidance (TSG), which improves structural plausibility and consistency by reducing disparities between attention maps across decoder blocks, and Fourier-based Motion Enhancement (FME), which amplifies motion by increasing the energy in the attention maps through frequency manipulation. Experimental results demonstrate BroadWay's effectiveness across multiple T2V backbones, showing substantial improvements in video quality, structural consistency, and motion richness without additional training or memory cost.

**Strengths:**

1. The proposed method offers a practical advantage by boosting T2V model performance without retraining, making it suitable for integrating with existing models.

2. The TSG and FME components are well-designed to tackle specific weaknesses in T2V generation, providing both structural coherence and dynamic motion in synthesized videos.

3. The authors validate their methods on multiple backbones like AnimateDiff and VideoCrafter2 illustrating its generalizability across various models, including potential applications in image-to-video tasks.

**Weaknesses:**

1. The hyper-parameters, especially \alpha (for TSG) and \beta (for FME), may need to be manually tuned for different backbones, reducing the plug-and-play convenience for some users. For example, the \beta parameter varies greatly for different T2V base models (1.5 for AnimateDiff and 10.0 for VideoCrafter2). Is there any guidance for this? The author should provide a sensitivity analysis or guidelines for selecting these parameters across different backbones. This would help users more easily apply the method to various models.

2. All the T2V backbones are based on the U-Net structure with an interleaved spatial-temporal attention module. How about the model with DiT, as well as full 3D attention? It is encouraged to discuss the potential of the proposed method when applying to models with different architectures like DiT or full 3D attention.

3. It is highly encouraged to use VBench to evaluate the proposed method since this metric could evaluate the jittery and motion magnitude of the videos.

4. UNet should be U-Net.

**Questions:**

Please refer to the weaknesses part.

---

### Note · Authors · 2024-11-13

**Comment:**

Thanks for your comments, we decide to make a thorough revision and resubmit this manuscript.

**Withdrawal Confirmation:**

I have read and agree with the venue's withdrawal policy on behalf of myself and my co-authors.